# In Silico Genomic Analysis of Avian Influenza Viruses Isolated From Marine Seal Colonies

**DOI:** 10.3390/pathogens13111009

**Published:** 2024-11-16

**Authors:** Klaudia Chrzastek, Darrell R. Kapczynski

**Affiliations:** 1Exotic and Emerging Avian Diseases Research Unit, U.S. National Poultry Research Center, Agricultural Research Service, U.S. Department of Agriculture (USDA) 934 College Station Road, Athens, GA 30605, USA; 2Center for Translational Antiviral Research, Institute for Biomedical Sciences, Georgia State University, Atlanta, GA 30303, USA

**Keywords:** avian influenza virus, marine mammals, seals, LPAIV, HPAIV, H5N1

## Abstract

Genetically diverse avian influenza viruses (AIVs) are maintained in wild aquatic birds with increasingly frequent spillover into mammals, yet these represent a small proportion of the overall detections. The isolation of AIVs in marine mammals, including seals, has been reported sporadically over the last 45 years. Prior to 2016, all reports of AIVs detected in seals were of low-pathogenicity AIVs. In spite of this, the majority of reported AIV outbreaks caused fatal respiratory diseases, with harbor seals particularly susceptible to infection. The H5 clade 2.3.4.4b highly pathogenic AIV (HPAIV) was detected in seals for the first time in 2016. Recently, many cases of mass seal die-offs have occurred because of 2.3.4.4b HPAIV and are attributed to spillover from wild bird species. The potential for seal-to-seal transmission has been considered after the mass mortality of southern elephant seals off the coast of Argentina. Close contact between seals and wild birds, the rapid evolution of H5N1 AIVs, and the possibility of efficient mammal-to-mammal transmission are increasing concerns due to the potential for the establishment of a marine mammal reservoir and public health risks associated with the pandemic potential of the virus. This manuscript details the detection of AIVs in the seal population, comparing interesting features of various subtypes with an emphasis on avian-to-mammal-to-mammal transmission. Phylogenetic characterizations of the representative seal isolates were performed to demonstrate the relationships within the different virus isolates. Furthermore, we demonstrate that the reassortment events between different LPAIVs occurred before and after the viruses reached the seal population. The reassortment of viral segments plays an important role in the evolution of influenza viruses. Taken together, these data report on the 45 year history between seals and AIVs.

## 1. Introduction

Avian influenza viruses (AIVs) are classified as either low-pathogenicity (LP) or high-pathogenicity (HP) based on the presence of basic amino acids at the hemagglutinin (HA) protein cleavage site and virus-induced mortality in chickens. LPAIVs generally cause mild infections, while HPAIVs can result in high mortality in a wide range of avian species. AIV subtypes are defined based on their surface glycoproteins, HAs (H1–H16) and neuraminidases (NA; N1–N9), with HPAIVs restricted to only the H5 and H7 subtypes. These viruses can be isolated from many animal species, including marine mammals. Until 2016, no HPAIV was reported in seal populations. Currently circulating strains of H5 HPAIV emerged in 1996 from southern China as A/goose/Guangdong/96-lineage (GsGD) and then evolved rapidly into 10 separate clades (0–9), expanding the geographical locations and animal species infected. The cross-species transmission of current H5N1 HPAIV clade 2.3.4.4b has been reported in over 43 mammalian species across Europe, North America, South America, and Asia [1]. Seals are one of the marine mammal species massively impacted by the current H5N1 HPAIV outbreak. In this manuscript, we detail the detection of AIVs in seal populations. We performed phylogenetic analysis of the seal isolates obtained over the last 45 years. In addition, we analyzed 72 PB2 proteins from different AIV isolates (seals and other marine mammals), available from the GISAID, and focused on two well-known adaptive mutations (D701N and E627K) that enhance replication of AIV in mammalian cells [2].

Seals, along with walruses and sea lions, are classified as pinnipeds (suborder Pinnipedia). There are 32 species of seals that belong to 17 genera and 2 families (Otariidae and Phocidae). Harbor seals (*Phoca vitulina*) belonging to the genus *Phoca* (common seals) represent Northern Hemisphere seals, and they are the most frequently AIV-infected seal species. They are nonmigratory, earless seals that can be found along coastlines and in a few freshwater lakes in Canada and Alaska [3] Gray seals (*Halichoerus grypus*) are only found in the North Atlantic, and there are three populations that are recognized, as follows: Northeast Atlantic, Northwest Atlantic, and Baltic Sea [4]. A large colony of gray seals can be found in the UK [5], with an estimated population of more than 120,000 animals, representing 40% of the world’s population and 95% of the European population of gray seals ([6]; Figure 1). In USA waters, the number of pupping sites has increased from one in 1988 to nine in 2019 and are located in Maine and Massachusetts [7]. From 1988 to 2019, the number of pups born has increased; however, these numbers do not reflect changes in total population growth resulting from Canadian seals migrating to the regions in the USA [7].The other seal of interest representing the Northern Hemisphere is the northern elephant seal (*Mirounga angustirostris*), with the largest colonies in southern California located in the Channel Islands. They have one of the longest migrations of any mammal; some have been recorded traveling over 13,000 miles roundtrip. These seals were nearly extinct in 1892 [8] but have increased in number over the last 70–80 years [9,10]. These seals were positive for pandemic H1N1 influenza virus twice (in 2011 and 2019) in the absence of clinical signs of disease. In contrast, the southern elephant seal (*M. leonina*) can be found throughout sub-Antarctic regions and have a greater body mass than their northern counterpart. They occasionally can be found in Antarctica to breed, but breeding locations are also distributed in sub-Antarctic locations. The largest seal breeding population is in the South Atlantic on South Georgia [11], with other breeding locations including the Falkland Islands and Valdes Peninsula in Argentina. Two big populations of southern elephant seals can be found in the South Indian Ocean, mainly on the Kerguelen Islands, and the sub-Antarctic islands of the Pacific Ocean, south of Tasmania and New Zealand, and around Macquarie Island. Recent reports demonstrate that the clade 2.3.4.4b H5N1 virus caused massive mortality in southern elephant young pup seals that were born in 2023 across Argentina, where 96% of the total population were dead after viral infection [12].

## 2. Low-Pathogenic Avian Influenza Virus (LPAIV) Infection in Seals

General information on LPAIVs found in seal populations is shown in Table 1. The first cases of AIVs that manifested as a clinical disease (pneumonia with a fatal outcome) in seals were of the low-pathogenic genotype. In the winter of 1979–1980, approximately 600 harbor seals (*Phoca vitulina*) presented clinical pneumonia that ended in severe mortality on Cape Cod peninsula in the USA [13,14,15]. The LPAIV H7N7 (previously called serotype Hav1 Neq1) was confirmed in lung and brain tissues [13]. Interestingly, this virus was harmless to chickens and turkeys in transmission experiments but previous human infection by H7N7 (Hav1 Neq1) was reported [13]. Similarly, Webster et al. [14] demonstrated that the seal H7N7 viruses replicated poorly, produced no disease signs, and were not shed in the feces of avian species, yet they were able to replicate in other mammals, namely, ferrets, cats, and pigs with no clinical disease. Antibodies to this virus were not detected in harbor seals but were found in gray and fur seals along the New England coast, suggesting that this virus may be newly introduced to this species [14]. The A/Seal/Mass/1/80 H7N7 influenza virus isolate provides the first evidence suggesting that a strain deriving all of its genes from one or more avian influenza viruses can be associated with severe disease in a mammalian population in nature [14].

Hinshaw et al. [16] demonstrated that the H7N7 seal isolate replicates in ferrets and reaches a peak titer of 10^8^EID50 mL within two days post-infection. This virus was also recovered from pigs and squirrel monkeys [16,17]. Later, Li et al. [18] demonstrated that this H7N7 virus can adapt to chicken embryo cells (CECs) by introducing a mutation close to the HA cleavage site, which resulted in cleavability by ubiquitous proteases and enhanced pathogenicity in chickens. In contrast, adaptation to MDCK cells led to mutations that were distant from the cleavage side, but the virus was still apathogenic for chickens [18]. The second time harbor seals were reported dying of pneumonia along the New England coast was from June 1982 to March 1983 [19]. After performing serological and RNA-RNA assays, it was demonstrated that these viruses were antigenically and genetically related to circulating AIV strains but distinct from mammalian influenza viruses, including H7N7 isolates recovered from seals in 1980, and were later classified as the H4N5 subtype (A/seals/Mass/133/82) [19] Interestingly, these seal viruses were able to replicate in the intestinal tracts of ducks compared to the previous isolate (H7N7), which replicated very poorly or not at all [19]. Experimentally infected with H4N5, harbor seals (N = 1), ringed seals (N = 2), and harp seals (N = 3) demonstrated mild clinical signs of disease. The virus was recovered from nasal (5/6) and anal swab samples (2/6). The seals also produced antibodies to the challenge virus (3/3) at two weeks post-challenge, which were still present at 30 days post-challenge [19]. This H4N5 virus replicated to high titer in lungs (10^5.5^EID50) compared to 10^3.5^EID50 in brain and lymph nodes tissues. No previous evidence of contact with this virus was observed in the seal population before 1982 [19]. We performed a phylogenetic analysis of the first seal isolate, A/seal/Massachussetts/133/1982 H4N5, and compared it with all other wild bird isolates available in the GISAID database. For the maximum likelihood (ML) phylogenetic analysis, all full-length HA segments of H4Nx bird isolates available in the GISAID (n = 3023) were retrieved, and representative sequences (n = 189) were selected based on the sequence identity at the 97% level using the CD-HIT package [20]. The percentage of identity between the H4N5 seal and representative wild bird isolates is shown in Appendix A. Only the isolates (n = 70) that clustered with the H4N5 seal isolate are shown in Figure 2. Based on our phylogenetic analysis, the first seal isolate, A/seal/Massachussetts/133/1982 (H4N5), clustered with viruses isolated from wild birds in North America before 1982. However, the similarity in the HA segment among them does not exceed 92% (Appendix A). The phylogenetic analysis of the HA suggests that the seal H4N5 LPAIV from 1982 might have been a reassortant AIV. This event most probably occurred naturally within the wild bird population and allowed the virus to appear later in seals. Furthermore, two H4N7 isolates found in wild birds in Alaska (in 2006 and 2009) clustered, along with a seal H4N5 (Figure 2), suggesting that reassortment events also occurred naturally afterwards.

Following these two epizootics of AIVs in seals in the 1980s, the prevalence of the influenza virus on Ross Island, in the Antarctic region, was examined, using samples collected in November 1979 from Weddell seals, Adelie penguins (N = 100), and Antarctic skuas (N = 60), in addition to serum samples collected from October 1985 to January 1986 [21]. No influenza virus was isolated, and none of the seal sera were positive for AIVs. However, the antibodies against AIVs were detected (most probably of the H10 serotype) in penguins and skuas from the Ross Sea Dependency, Antarctic region [21].

In January 1991 and from January to February 1992, harbor seals were found dead along the Cape Cod peninsula of Massachusetts, and two AIVs were identified at that time—H4N6 and H3N3—of which, H3N3 was isolated for the first time in the seals [22]. The antigenic reactivity of H3 viruses isolated from seals demonstrates that two seal viruses (A/Seal/MA/3911/92 and A/Seal/MA/4007/92) had similar genetic patterns to A/Duck/Ukraine/1/63, and one virus (A/Seal/MA/3984/92) had a pattern more similar to the human virus, A/Aichi/2/68 [22]. Genetically the HAs of these new seal viruses were closely related to the HAs of viruses recovered from North American birds [22]. Callan et al. [22] demonstrated that residues 226 and 228 in the HAs of the seal isolates are glutamine and glycine, respectively, which coincides with the consensus for the receptor-binding site of most avian and equine H3 viruses. The H7N7 (A/Seal/Mass/1/80) and the H4N5 (A/Seal/MA/133/82) viruses previously isolated from seals also have the same avian receptor-binding sequence, G225-Q-S-G-R229 [23,24]. This suggests that the binding sequence may be well adapted for seals, since mutations away from this sequence have not been observed in viruses isolated from seals [22]. Here, we performed a phylogenetic analysis of complete H3N3, H3Nx, and HxN3 segments of AIV isolates available in the GISAID database. Full-length HA segments of the H3Nx seal and wild bird isolates (n = 1840) were retrieved, and representative sequences (n = 256) were selected based on the sequence identity at the 97% level using the CD-HIT package [20]. All isolates that clustered with a seal isolate (n = 38), which represented one cluster, were selected to generate the final phylogenetic tree (Appendix A). As expected, all H3N3 seal isolates (A/seal/MA/3984/1992, A/seal/MA/3911/1992, and A/seal/Massachusetts/3911/1992) clustered together. Based on the HA analysis, their precursors were avian influenza viruses from the USA (A/mallard_duck/New_York/157/1986 H3N6, A/mallard/Ohio/48/1986 H3N2, and A/mallard/Ohio/264/1986 H3N8), which shared 98–98.8% similarity to the seal isolates (Table 1, Appendix A, Appendix A). For the NA analysis, the representative sequences (n = 181) were selected based on the sequence identity at the 97% level, using the CD-HIT package, out of all HxN3 isolates available in the GISAID database. The isolates that clustered with A/seal/Massachusetts/3911/1992 H3N3 were selected (n = 51) for further analysis (Appendix A). The NA segment of A/seal/Massachusetts/3911/1992 (H3N3) shared an approximately 99.4% sequence identity with the viruses H2N3, H4N3, and H7N3, which were circulating among wild birds in Canada (Alberta) in 1988–1990 (Table 1, Appendix A, Appendix A). The high similarity of the seal H3N3 to different virus subtypes suggests that reassortment occurred even before the viruses reached the seal population. Matrosovich et al. [25] studied four H3N3 viruses isolated from the seals during the outbreak and assessed their affinity for sialylglycopolymers (analogs of cellular receptors) 3′SL-PAA (avian receptor) and 6′SLN-PAA (human receptor). Three out of four isolates had the typical avian-virus-like pattern of binding to sialylglycopolymers; the apparent association constants with 3′SL-PAA and 6′SL-PAA were about 50 μM−1 and 1 μM−1, respectively. This finding suggests that these avian influenza viruses can infect seals without substantial changes in their receptor-binding specificity. One of the isolates had a lower binding affinity for 3′SL-PAA without an increase in binding to 6′SLN-PAA, and it carried two amino acid substitutions, A138S and R220S [25]. Recently, Ramis et al. [26] assessed the patterns of attachment for different influenza virus strains (H4N5, H7N7, and human H1N1, H3N2, and Human B influenza) in harbor seals, gray seals (*Halichoerus grypus*), harbor porpoises (*Phocoena phocoena*), and bottlenose dolphins (*Tursiops truncatus*). It was demonstrated that the attachment of AIVs to tracheal and bronchial epithelia was moderate in seals and absent in the harbor porpoise and bottlenose dolphin, which suggests that seals are susceptible to these viruses and transmission of them may occur. Lack of attachment to trachea and bronchi of harbor porpoises and bottlenose dolphins suggests low susceptibility and inefficient transmission in these species [26]. The H4N6 virus was also isolated from the Caspian seal at Zhemchuzhny island (Astrakhan Region, Russia) in 2012; the seal did not present clinical signs of disease [27]. An approx. 98% similarity of A/Caspian_seal/Russia/T1/2012 to A/duck/South Africa/1233A/2004 H4N8 and A/duck/Nanjing/1102/2010 H4N8 was observed based on the HA analysis, whereas the NA segment shared 98.8% with A/mallard duck/Netherlands/7/2006 H10N6 (Table 1). In addition, no mammalian adaptational mutations were found in PB2 protein [27]. The virus replicated in the lungs of mice and caused severe disease without prior adaptation [27]. Serological evidence for the infection of Caspian seals with the influenza A and B viruses was demonstrated using the sera collected in 1993–2000, which reacted strongly with the A/Bangkok/1/79 (H3N2) strain, suggesting that the seals were infected with the human influenza A virus that circulated in 1979–1981 but did not react at all with the A/Philippines/2/82 strain, which was prevalent in humans in 1982–1983 [28].The youngest seal with positive serum was estimated to be 14.5 years old in 2000, implying that A/Bangkok/1/79-like viruses were maintained in the Caspian seal population until at least 1985 [28]. However, in 2012, none of 27 Caspian seal sera samples tested positive against H1, H3, H5, and H7 AIVs [27].

### 2.1. LPAIV: H3N8 (2011, 2017–2019)

In September 2011, a new outbreak of AIV was reported along the New England coast in the USA, where harbor seals (*Phoca vitulina*) were found dead or moribund. This virus was identified as the H3N8 subtype and had never been isolated from seals prior to this event; however, it was detected in harp seals in the Northwest Atlantic Ocean [29,30]. An “unusual mortality event” was also declared in Maine, New Hampshire, and northern Massachusetts in November 2011 [31]. Genetic characterization of this virus demonstrated similarity to a waterfowl isolate circulating in North America since at least 2002 [30]. Based on the HA analysis performed here, A/harbor_seal/New_Hampshire/M11t72H3/2018 H3N8 and A/seal/Alaska/PV2114/2021 H3N6 clustered with A/mallard/Oregon/AH0038704/2015 H3N8 (Appendix A), with genetic distances of 0.0382 and 0.0325, respectively (Appendix A). Interestingly, this new seal H3N8 isolate demonstrated an adaptive mutation, D701N, in the viral PB2 protein; however, it maintained the avian phenotype at positions 226 (Q) and 228 (G) in the HA protein, which correlates with a preferential ability to use the SAα-2,3-avian-type receptor [30]. Later, the binding properties and preferences of the seal H3N8 virus to α2-3-linked and α2-6-linked sialosides were investigated in more detail [31,32,33]. Karlsson et al. [32] demonstrated that seal-origin H3N8 is unique among the avian H3N8 viruses tested, with enhanced α2,6 receptor binding, increased morbidity in mice, and efficient respiratory droplet transmission in ferrets, which was not observed for the other avian viruses within this clade. Yang et al. [31] performed a detailed structural and biochemical analysis of the surface antigens of the A/harbor seal/New Hampshire/179629/2011 virus and demonstrated that both the HA and NA indicate a true avian receptor-binding preference with strong binding affinity to the α2-3-linked sialosides, as well as mixed α2-3/α2-6 branched sialosides (glycans 65 to 66) but only slight binding to human α2-6-linked sialosides. Hussein et al. [33] evaluated the potential human transmissibility of seal H3N8. The binding of the recombinant HA proteins of seal H3N8 and human-adapted H3N2 viruses to respiratory tissues of humans and ferrets was tested, demonstrating that there was a strong tendency of the seal H3 to bind to lung alveoli in human tissue, which is opposite to the human-adapted H3 that bound mainly to the trachea [33]. Furthermore, the binding of the recombinant HAs to a library of 610 glycans demonstrated that the seal H3 bound preferentially to α-2,3-sialylated glycans, which is in contrast to the human H3, which bound almost exclusively to α-2,6-sialylated glycans. These results suggest that seal H3N8 virus has retained its avian-like receptor-binding specificity but could potentially establish infection in human lungs [33]. Avian-lineage H3N8 virus was next found in a gray seal pup in Cornwall, UK, in 2017. The pup underwent medical treatment but died after two weeks [34]. Based on a BLAST search (Basic Local Alignment Search Tool) and phylogenetic analysis, this virus most probably originated from unsampled, locally circulating (in Northern Europe) viruses, likely from wild Anseriformes. Furthermore, several mutations were detected, including D701N in the PB2 segment, which is a rare mutation in this virus subtype, and suggest a mammalian adaptation of bird viruses [34].

Interestingly, the first human case of H3N8 AIV infection was confirmed in China in 2022 [35,36]. The person infected developed severe pneumonia. Wang et al. [37] have shown that H3N8 viruses circulating in wild migratory birds in China have undergone complicated reassortment with viruses in waterfowl; they preferentially bind to the avian-type receptor, similar to the H3N8 found in seals [33], but they acquired the ability to bind to human-type receptors.

### 2.2. LPAIV: H10N7 (2014–2015 and 2021)

In 2014, mass mortality of harbor seals (*Phoca vitulina*) occurred in Sweden (March 2014), Denmark (July 2014), and Germany (October 2014), where the seals were found dead and washed up on the shores [38,39,40]. Those were the first ever reported AIV cases of dead seals outside of the USA. From March through October 2014, 425 carcasses were detected in several seal colonies in the Kattegat and the Skagerrak Seas in Sweden [38]. The H10N7 subtype was detected and based on HA characterization of the virus clustered with seal isolates from Germany (2014) and a Swedish mallard H10 isolate from 2011 [38]. In Denmark, 152 harbor seals on the Island of Anholt were found dead from severe pneumonia between June and August 2014 [40]. In general, both the HA and NA segments showed a high-level nucleotide sequence identity to AIVs from birds sampled in Scandinavia and the Republic of Georgia [40]. In Germany, dead seals accounted for 12% of the whole population which at the time was approximately 12,000 animals in total [41]. Influenza A virus (A/harbor seal/Germany/1/2014) of the H10N7 subtype was isolated from lung and throat swab samples and replicated to a high level in 11-day-old embryonated chicken eggs and MDCK cells. Genetic characterization of the virus showed that seal H10N7 is closely related to the H10N7 viruses found in migratory ducks in Georgia, Egypt, and the Netherlands in 2009–2014 [40]. Interestingly, at the amino acid level, the HAs of the viruses from Denmark and Germany were 99.3% identical, and the very first Swedish isolate was approximately 97.5% identical to all other strains [40]. This lower amino acid identity of the HAs between Swedish and other H10N7 viruses is reflected by nucleotide mutations (dN/dS = 0.7), indicating that adaptation to seals was in progress [40]. A limited number of seals (<180) were also found dead in the Netherlands from early November 2014 until early January 2015 (Bodewes et al. 2016). A serological investigation showed that antibodies against H10N7 was found in 41% (32 out of 78) of pups, 10% (5 out of 52) of weaners, and 58% (7 out of 12) of subadults or adults harbor seals in 2015 [41]. Although no cases of dead gray seals were reported, 26% (5 out of 19) were seropositive to influenza virus in 2015 [41]. These findings indicate that, despite apparent low mortality, infection with seal influenza A(H10N7) virus was geographically widespread and also occurred in gray seals [41]. Brand et al. [42] have shown that the A/seal/Germany/2014 H10N7 virus causes respiratory disease in harbor seals and experimentally infected ferrets. The lesions in both species were restricted to the respiratory tract with no evidence of a spread to extra-respiratory tissues [42]. Viral antigen was predominantly found in bronchial and submucosal glandular epithelial cells in harbor seals, whereas in ferrets they were mainly found in bronchiolar and bronchial submucosal glandular epithelial cells and less frequently in type I and II pneumocytes, bronchial, bronchial glandular, and tracheal epithelial cells, which might suggest a slightly different cell-receptor distribution [42]. Bodewes et al. [41] have demonstrated that the majority of the sequence variations collected from seals during the course of the outbreak between April 2014 and January 2015 occurred in HA genes. The highest variation in the HA was observed at the beginning of the epidemic; afterward, the number of mutations observed earlier had been fixed, suggesting that when an AIV jumps a species barrier (from birds to seals), amino acid changes in the HA may occur rapidly, seeming to be important for a virus adaptation to its new mammalian host [41]. Dittrich et al. [43] utilized reverse genetics to generate recombinant avian H10N4 viruses that carried one of eight unique mutations or the complete wild-type HA from the seal virus. Wild-type recombinant H10N4 virus had high affinity to avian-type sialic acid receptors and no affinity to mammalian-type receptors. In contrast, Q220L (H10 numbering) in the rim of the receptor-binding pocket increased the affinity of the H10N4 virus to mammal-type receptors and completely abolished the affinity to avian-type receptors [43]. Furthermore, all viruses, including the wild-type H10N7 virus, replicated at higher levels in chicken cells than in human cells, suggesting that adaptive mutations (e.g., Q220L) enhanced replication in mammals and retained replication efficiency in the original avian host [43]. Herfst et al. [44] have shown that A/H10N7 viruses isolated from seals in Europe (A/harbor seal/NL/PV14–221_TS/2015 and A/harbor seal/S1047_14_L/Germany/2014) obtained changes in their HA segment that decreased the avidity of the virus for avian-type receptors and increased its preference for human-type receptors. Those changes were caused by substitutions in the 220-loop that forms one edge of the receptor-binding pocket, in particular the amino acid substitution Q226L which demonstrated stronger binding to the human-type sialic acid receptor [44]. Furthermore, these seal H10N7 viruses were aerosol or respiratory droplet-transmissible among ferrets [44]. Interestingly, Guan et al. [45] have demonstrated that H10N7 gull isolates found in Iceland, in 2015, were genetically related to the H10 that caused influenza outbreaks and deaths among European seals in 2014, which could be transmitted among ferrets through the direct contact and aerosol routes without prior adaptation.

The H10N7 virus was also isolated from dead harbor seal in British Columbia, Canada, in 2021, that demonstrated bronchointerstitial pneumonia [46]. This new H10N7 virus isolated in Canada is a reassortant virus between North America and Eurasia lineages [46]. Interestingly, this virus also carried a mutation at position 701 in the PB2 protein (D701N) that was not seen in the other H10N7 isolates from seals ([46], Appendix A). Most of the H10N7 viruses isolated from seals in Germany, the Netherlands, and Denmark clustered together; A/harbor_seal/Denmark/14-5061-1lu/2014-07 clustered with Swedish isolates (Appendix A). The percentage of similarity between the HAs of the H10N7 Canadian isolate from 2021 and the European H10N7 was between 90 and 92% (Appendix A). Furthermore, we also compared the HAs of H10N7 seal isolates with H10Nx wild bird isolates available in the GISAID database. The representative sequences of the HAs were retrieved based on the sequence identity at the 97% level using the CD-HIT package [20]. The genetic distances between the H10N7-seal origin and H10Nx wild bird isolates below 0.06 are shown in Appendix A. The A/Anas_platyrhynchos/Belgium/11798_H189445/2016 H10N5 and A/avian/Israel/543/2008 H10N7 were similar to the H10N7 European seal isolates but not to A/harbor_seal/British_Colombia/OTH-52-1/2021 H10N7 (Appendix A). The genetic distances between the A/little_curlew/Hebei/QHD716/2013 H10N7 and A/duck/Aichi/231110/2012 H10N8 and H10N7 seal isolates were between 0.037 and 0.056 (Appendix A).

The PB2 segment of the H10N7 Canadian seal isolate from 2021 shared an approximately 95% similarity with the 2.3.4.4b H5N1 seal, sea lion, and dolphin isolates obtained in Washington, USA, and marine animals (such as sea lion, dolphin, and porpoise) from South America. Remarkably, the Canadian seal H10N7 and the current H5N1 isolates from South America have the same D701N substitution in the PB2 protein (Appendix A). Similarly to PB2, the PB1 segment also clustered with the current H5N1 2.3.4.4b isolates (Figure 3 and Figure 4). In addition, the PB1 and MP segments clustered with the H3N8 isolates obtained from seals in the USA in 2011 and 2018 (Figure 4 and Figure 5). The PA segment clustered with the seal H4N5 isolate of 1982 from the USA but not with European H10N7 isolates (Figure 6). The phylogenetics analysis of the NP, HA, NA, and NS segments is shown in Appendix A, Appendix A, Appendix A, and Appendix A, respectively. The analysis of the HA segments from all seal isolates demonstrates that the hemagglutinins belonging to different subtypes are not closely related.

### 2.3. Pandemic H1N1 (2010, 2019)

In 2010, the pandemic of H1N1 (pH1N1) influenza A virus was first detected in marine mammals in northern elephant seals on the California coast. None of the seals displayed clinical signs of disease [47]. Genetic analysis demonstrated greater than 99% homology for all segments to pandemic influenza A/California/04/2009 that circulated in humans in California in 2009 [47]. The antibodies to pH1N1 were detected in seal populations in 2010 with subsequent virus isolation. Antibodies to pH1N1 were also detected in following years, 2011 and 2012, even though the PCR results were negative [47]. Interestingly, antibodies to pandemic H1N1 were also detected in pups born in 2011 even though the seals tested negative to the virus, which could suggest the passive transfer of maternal antibodies to the pups [47]. Between 2011 and 2018, surveillance of marine mammals along California’s coast was conducted, and the results showed frequent detections of antibodies against the influenza virus in seals but rarely PCR-positive virus detections [48]. In spring 2019, ten samples from northern elephant seals (*Mirounga angustirostris*) were PCR positive for pH1N1 avian influenza virus although the virus isolation was not successful. This represented the first report of human pandemic H1N1 in northern elephant seals since 2010, suggesting the influenza virus continues to spill over from humans to pinnipeds [48]. The HA of A/Elephant_seal/California/1/2010 shared 99.8% similarity with A/Singapore/GP3733/2009 H1N1, A/Malaysia/2082543/2009 H1N1, and A/Texas/JMS358/2009 H1N1, whereas the NA was most similar to A/Singapore/DMS7/2009 H1N1 and A/Scotland/Edinburgh_23151/2009 H1N1 (99.5%) (Table 1). These data supports the spillback of H1N1 from humans to seals for these viruses (Table 1).

## 3. Highly Pathogenic Avian Influenza Viruses (HPAIV)-Infected Seals

### 3.1. HPAIV: H5N8 Clade 2.3.4.4b (2016/2017, 2021, 2022)

Since the first report of AIVs in seals in 1979–1980s, it took over 30 years for HPAIV to be detected in seal populations and cause clinical disease. Two gray seals were found dead on the Baltic coast of Poland, the first one on November 27, 2016, and the second one 5 months later [49]. The lung samples were collected and the A/H5N8 clade 2.3.4.4 b virus was confirmed in both cases [49]. Only one virus was isolated, as isolation of the virus in the other animal failed; however, direct sequencing of the HA and NA genes was performed. This H5N8 seal virus contained 99.7%–100% sequence similarity to viruses that were circulating in aquatic wild bird species during the HPAIV outbreaks in 2016 and 2017. No mutations in the PB2 segment of the seal H5N8 were found. These first findings of HPAIV in seals suggested that cross-species transmissions can occur sporadically, and the possibility of seal-to-seal transmission should not be excluded [49].

A few years later, H5N8 HPAIV was detected in harbor and gray seals in the UK (2021), harbor seals in Germany (2021), and one harbor seal in Denmark (2022). An unusual disease event occurred at the Rehabilitation Center in the UK where different mammals, including red fox, gray seal, and several common seals were dead because of H5N8 HPAIV infection. The seals exhibited sudden-onset neurological signs, including seizures before death or euthanasia. These events occurred roughly 1 week after five swans housed in the same quarantine unit died from infections with the HPAI H5N8 virus [50]. Based on genetic and epidemiologic investigations, the swans were most likely the source of infection for the fox and seals; virus transmission likely occurred by fomite transfer or aerosol spread [50]. The D701N amino acid substitution in the PB2 gene was identified in both sequences derived from the mammalian species and was absent from all avian sequences generated during the 2020–2021 outbreak in the United Kingdom [50].

In mid-August 2021, three adult harbor seals (*Phoca vitulina*) were found dead along the German North Sea coast [51]. The following two variants of the H5N8 virus were detected from the seal samples: genotype Ger-10-20-N8, which dominated the avian epizootic 2020/2021 and was found in Germany from October 2020 until July 2021, and the second, clustered with genotype Ger-02-21-N8, a much rarer genotype that was only detected three times in Germany (in February and March 2021), with a novel NP segment sharing the highest identity with Eurasian LPAIV strains found in wild birds [51]. These findings suggest that there were most probably at least two independent entries of H5N8 into the seal population in Germany [51]. Some of these viruses demonstrated the E627K mutation in the PB2 segment (Appendix A).

### 3.2. HPAIV: H5N1 Clade 2.3.4.4b (2022–Current)

The current outbreak of 2.3.4.4b H5N1 viruses has affected over 43 mammalian species, including marine mammals, across Europe, North America, South America, and Asia [1]. The 2.3.4.4b H5N1 virus was reported in seals in North and South America (USA, Canada, Argentina, Chile, Uruguay, and Brazil) and Europe (Germany, Russia, Denmark, South Georgia, and the South Sandwich Islands, UK).

On 8 October 2023, the 2.3.4.4b H5N1 virus was detected for the first-time in the Antarctic and sub-Antarctic regions of South Georgia and the Falkland Islands (British Overseas Territory of South Georgia at Bird Island) [52]. The main infected bird species was brown skuas, for which the mortality rate increased rapidly over the month. The virus was confirmed in skuas and kelp gulls across four different sampling locations in South Georgia and southern fulmar in the Falkland Islands [52]. Clinical disease also manifested in elephant and fur seals in South Georgia [52]. Genetic assessment of the virus indicates the spread from South America, likely through the movement of migratory birds [52].

In South America, Argentina reported the mass mortality of young pups born in 2023 across the country, which represented almost 96% of all southern elephant (*Mirounga leonina*) seal pups born across Argentina [12] Previously, a mass mortality event of more than 3000 sea lions (*Otaria flavescens*) was observed in January and February 2023 in Peru [53]. These viruses belong to the HPAI A/H5N1 lineage 2.3.4.4b and are 4:4 reassortants, where PA, HA, NA, and MP belong to a Eurasian lineage that initially entered North America from Eurasia, and the remaining PB2, PB1, NP, and NS came from an American lineage that was already circulating in North America [54]. As reported by Leguia et al. [54], these viruses do not contain mutations linked to mammalian host adaptation and enhanced transmission (such as PB2 E627K or D701N), but at least eight novel polymorphic sites were found in their genome. Detection of the H5 virus in sea lions (*Otaria flavescens*) was also reported in Chile on 10 February 2023 [55]. The genetic characterization of isolates obtained from birds and marine mammals viruses revealed that all Chilean H5N1 viruses belong to the lineage 2.3.4.4b and cluster monophyletically with viruses from Peru, indicating a single introduction from North America into Peru/Chile [56]. Both D701N (in two sea lions, one human, and one shorebird) and Q591K (human and one sea lion) mutations were identified in the PB2 segments [56]. Interestingly, a minor population of viruses carrying the D701N mutation was present in 52.9%–70.9% of the sequence reads obtained from the samples tested, suggesting a mixed population of viruses within the sample [56].

Previously, HPAIV H5N1 infection was reported in seals in the USA (New England). Sequencing of 71 avian- and 13 seal-derived virus genomes from New England demonstrated all but 1 virus represented non-reassortant Eurasia 2.3.4.4b viruses [57]. The authors concluded that viral outbreak among New England harbor and gray seals was concurrent with a wave of avian infections in the region, and evidence of mammal adaptation existed in a small subset of the seals (PB2 E627K or D701N mutations) [58]. Many years before the HPAIV outbreak, Puryear et al. [58] demonstrated that North Atlantic gray seals from Cape Cod, MA, USA, and Nova Scotia, Canada, were consistent for AIVinfection and that serum demonstrated a broad reactivity to diverse influenza subtypes suggesting that seals might possibly represent an endemically infected wild reservoir of AIVs.

Analysis was performed on PB2 proteins from 72 seal and marine mammal isolates available in GISAID (58 seals isolates of different subtype isolated between 1980 and 2023 along with recently isolated H5N1 clade 2.3.4.4b viruses from 8 sea lions, 3 dolphins and 3 porpoises) (Appendix A). A mutation D701N was found in H3N8 seal isolates isolated in 2011 and 2018 in the USA and H10N7 seal isolates obtained from Canada in 2021. However, no mutation D701N was found in H10N7 from 2014 and 2015 isolated during epizootic in Europe. Several mutations D701N in PB2 were also found in recent 2.3.4.4b H5N8 and H5N1 seals isolates, as well as sea lions, dolphins, and porpoises isolated from South America in 2023 (Appendix A). Interestingly, none of the LPAIV seal isolates carried the E627K mutation in PB2, and this mutation was only found in some of the H5Nx viruses in the 2.3.4.4b clade that were recently isolated (e.g., H5N8 from Germany and Denmark and H5N1 from Scotland, USA) (Appendix A).

## 4. Summary

In this report, we briefly analyzed 45 years of AIV history in seal populations. The very first cases of AIVs were reported in harbor seal populations (*Phoca vitulina*) in the late 1970s, with subsequent detections in gray seals (*Halichoerus grypus*). All AIVs detected between 1979 and 2016/17 were of a low-pathogenic pathotype (LPAIV). Some LPAIVs demonstrated features that allowed them to adapt to a new species (such as mutation in the PB2 segment), and they are able to replicate in the seal population with or without clinical signs of disease. This was later confirmed by seroprevalence and virus isolation in apparently healthy animals. The H5N8 HPAIV clade 2.3.4.4b was first detected in a seal population in 2016/17. The 2.3.4.4b H5N1 virus is currently causing the mass mortality of seals and sea lions in the Americas, drastically reducing their populations. The H5N1 virus reached the Antarctic region, which is an important breeding ground for many key species. This event is alarming and will have ecological repercussions in seabirds, penguins, and other marine mammals in this remote region. Evidence of influenza A viruses in various animals has been reported in Arctic regions, with 26 unique low and highly pathogenic subtypes having been characterized in the scientific literature [59]. The work cited here provides characterizations of the interspecies transmissions of AIVs between birds and marine mammals and demonstrates that seals might be an important wild reservoir of influenza that contributes to the mammalian adaptation of these viruses. This, along with the rapid evolution of H5N1 AIV, the virus’s ability to reassort with other influenza viruses, particularly in high-risk settings, such as farms or areas highly crowded with animals, and the possibility of efficient mammal-to-mammal transmissions raises concerns about the potential for more transmissible variants to arise and thus public health risks associated with a potential pandemic.

## 5. Methods

In this review, we performed multiple phylogenetic analyses of different AIV isolates obtained from seals. The nucleotides sequences were aligned using MUSCLE and MEGA 11.03.13 software. The estimates of the evolutionary divergence among the sequences used for the data analyses were conducted using the maximum composite likelihood model and MEGA 11.03.13 software. The phylogenetic trees were created using a GTR nucleotide substitution model, with an among-site rate variation model using a discrete gamma distribution. Bootstrap support values were generated using 500 rapid bootstrap replicates.

## Figures and Tables

**Figure 1 pathogens-13-01009-f001:**
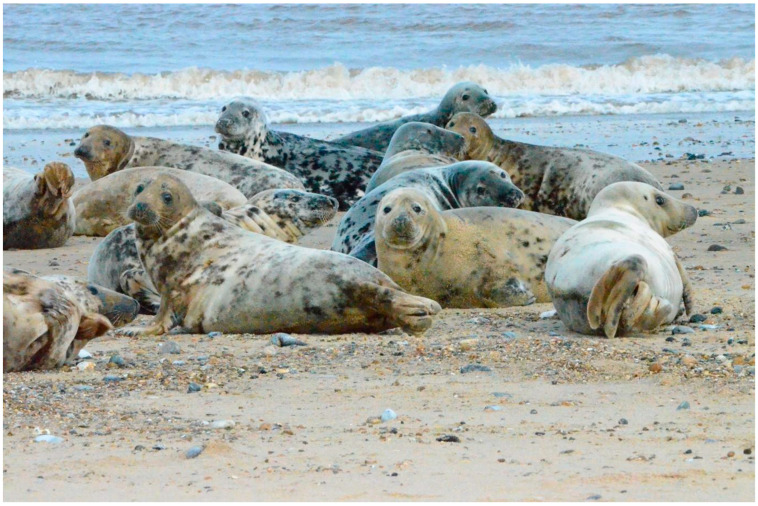
The Horsey gray seals colony, United Kingdom, February 2024. Photography by KC.

**Figure 2 pathogens-13-01009-f002:**
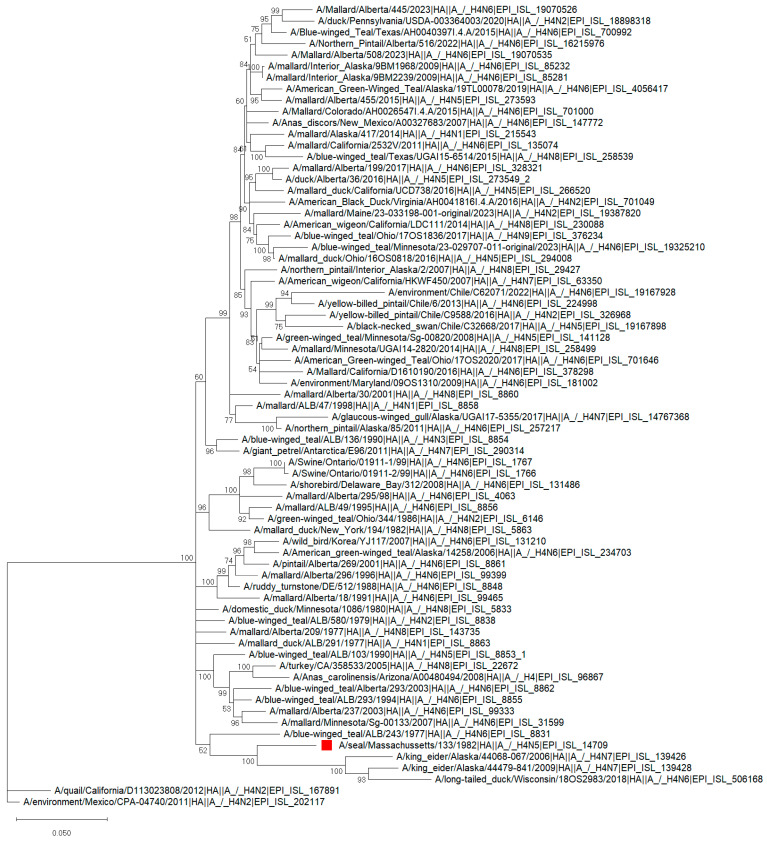
Phylogenetic tree of the HA segment of the H4N5 avian influenza virus isolate obtained from seals in 1982 (marked in red) and representative wild bird isolates available in the GISAID database. Full-length HA segments of H4Nx bird isolates available in the GISAID (n = 3023) were retrieved, and representative sequences (n = 189) were selected based on the sequence identity at the 97% level using the CD-HIT package. The isolates that clustered along with the H4N5 seal isolate (n = 70) were selected to construct the phylogenetic tree of the HA. The sequences were aligned using MUSCLE on MEGA 11.03.13. The GTR nucleotide substitution model, with an among-site rate variation model using a discrete gamma distribution, was used. Bootstrap support values were generated using 500 rapid bootstrap replicates.

**Figure 3 pathogens-13-01009-f003:**
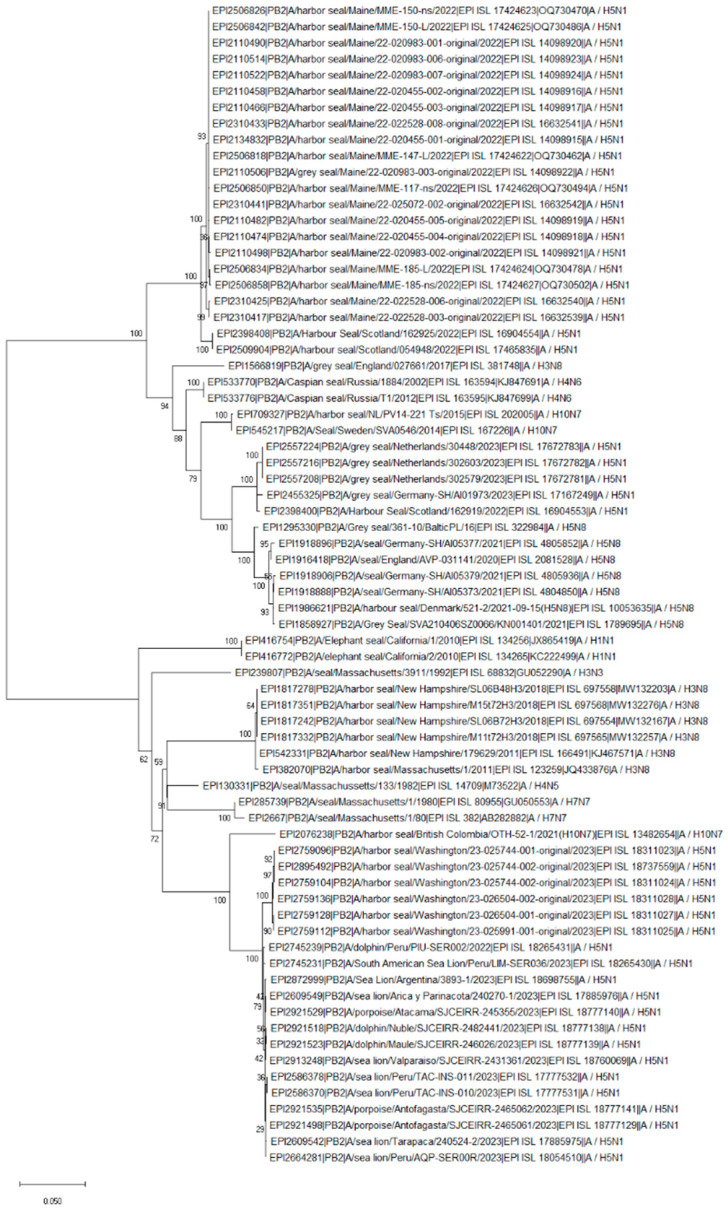
Phylogenetic tree of the PB2 segment of the avian influenza viruses found in seal populations over the last 45 years. The analysis includes the following 72 isolates available from GISAID (as of March 2024): 58 seal isolates of different subtypes isolated between 1980 and 2023 and recently isolated H5N1 clade 2.3.4.4b viruses from 8 sea lions, 3 dolphins, and 3 porpoises. The nucleotide sequences of the PB2 segment were aligned using MUSCLE software and GTR nucleotide substitution model, with an among-site rate variation model using a discrete gamma distribution. Bootstrap support values were generated using 500 rapid bootstrap replicates.

**Figure 4 pathogens-13-01009-f004:**
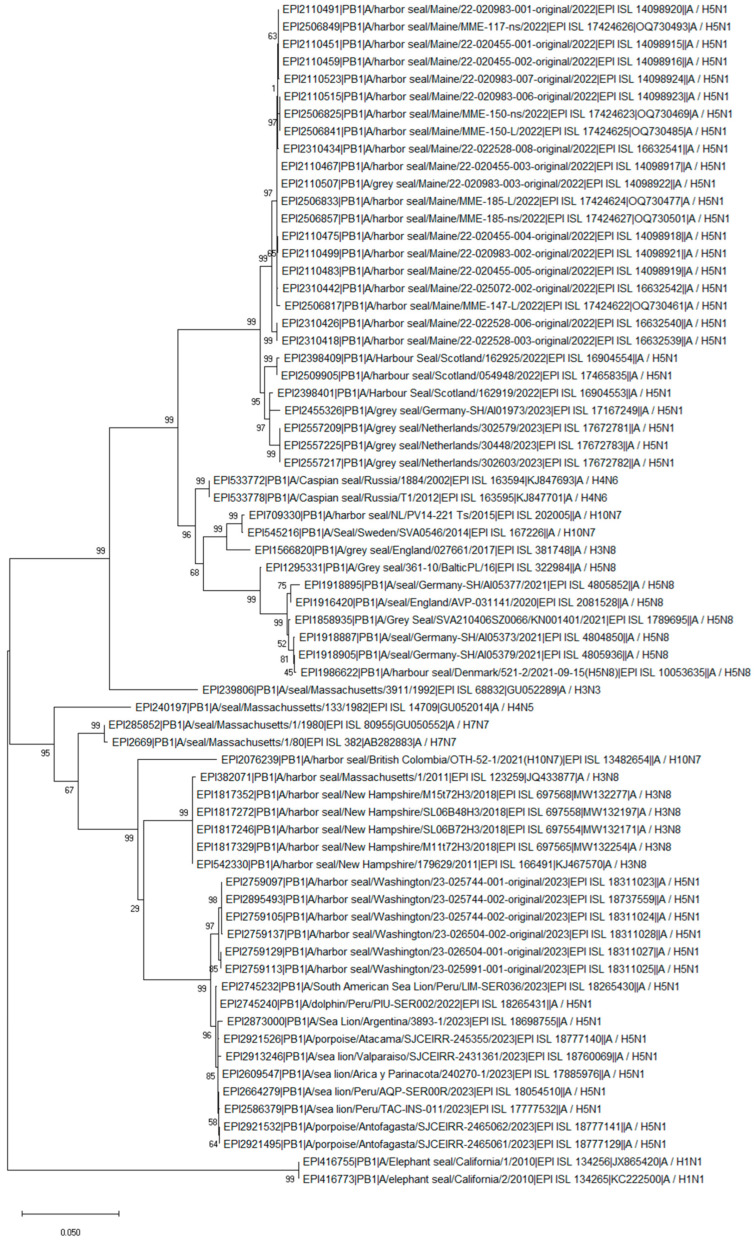
Phylogenetic tree of the PB1 segment of the avian influenza viruses found in seal populations over the last 45 years. The analysis includes PB1 segments that are available from GISAID (as of March 2024) and which were isolated between 1980 and 2023 and recently isolated H5N1 clade 2.3.4.4b viruses from 8 sea lions, 3 dolphins, and 3 porpoises. The nucleotide sequences of the PB1 segment were aligned using MUSCLE software and the GTR nucleotide substitution model, with an among-site rate variation model using a discrete gamma distribution. Bootstrap support values were generated using 500 rapid bootstrap replicates.

**Figure 5 pathogens-13-01009-f005:**
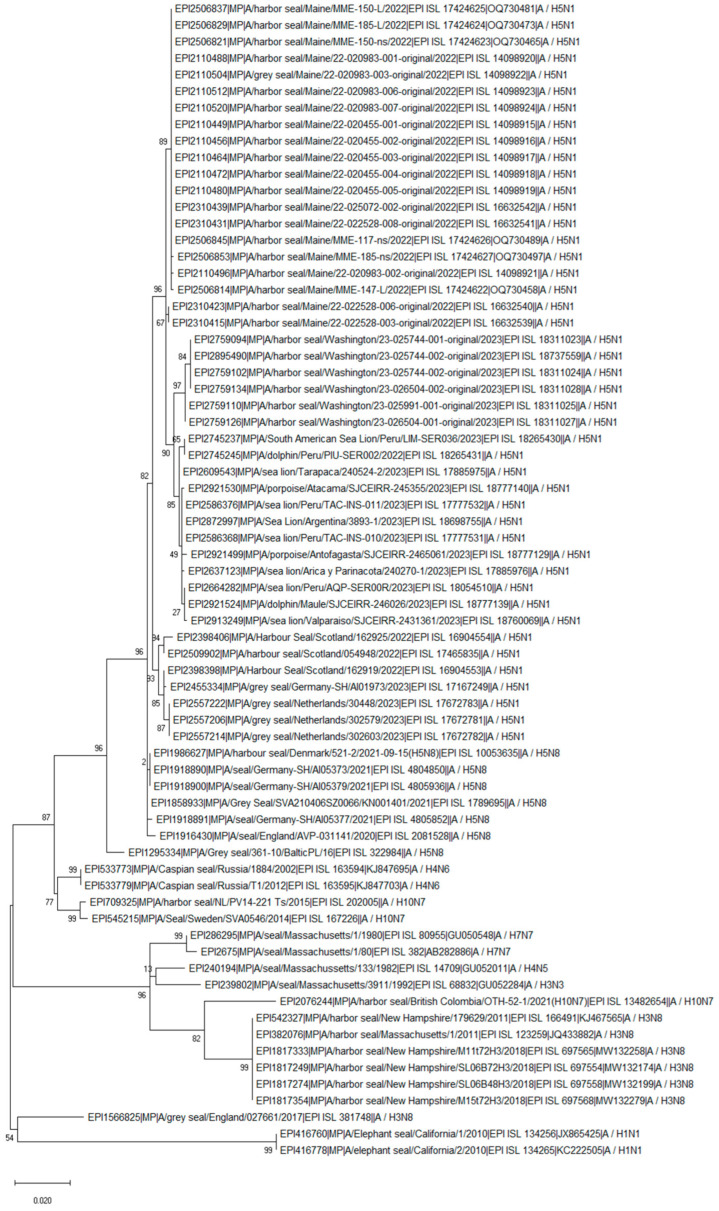
Phylogenetic tree of the MP segment of the avian influenza viruses found in seal populations over the last 45 years. The analysis includes MP segments that are available from GISAID (as of March 2024) and that were isolated between 1980 and 2023 and recently isolated H5N1 clade 2.3.4.4b viruses from 8 sea lions, 3 dolphins, and 3 porpoises. The nucleotide sequences of the MP segment were aligned using MUSCLE software and the GTR nucleotide substitution model, with an among-site rate variation model using a discrete gamma distribution. Bootstrap support values were generated using 500 rapid bootstrap replicates.

**Figure 6 pathogens-13-01009-f006:**
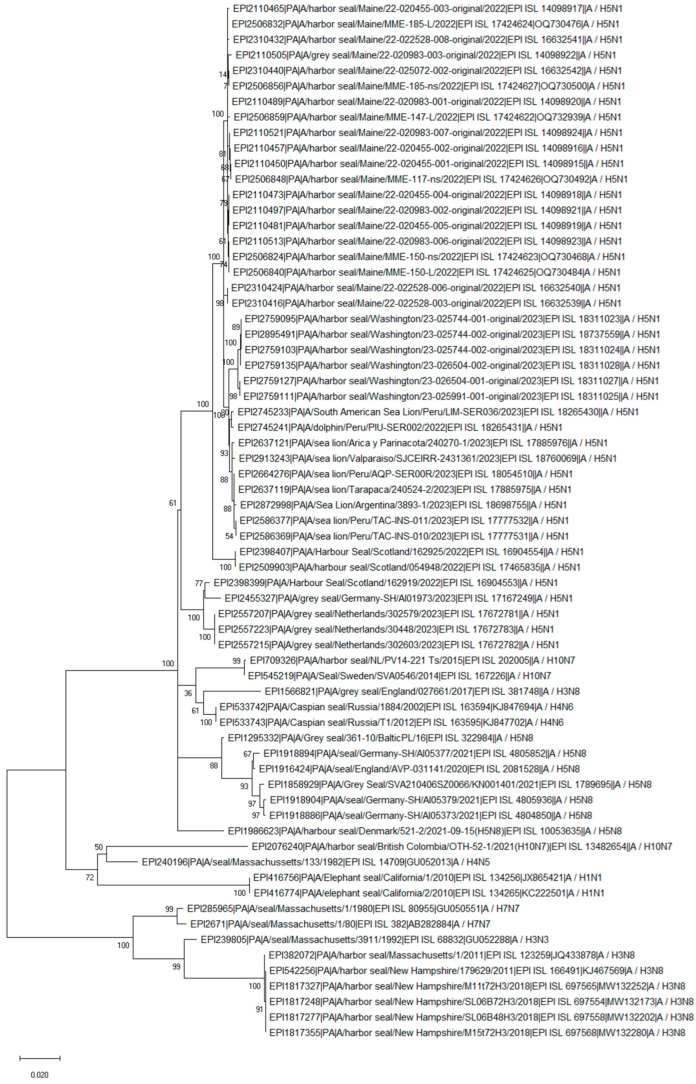
Phylogenetic tree of the PA segment of the avian influenza viruses found in seal populations over the last 45 years. The analysis includes PA segments that are available from GISAID (as of March 2024) that were isolated between 1980 and 2023 and recently isolated H5N1 clade 2.3.4.4b viruses from 8 sea lions, 3 dolphins, and 3 porpoises. The nucleotide sequences of the PA segment were aligned using MUSCLE software and the GTR nucleotide substitution model, with an among-site rate variation model using a discrete gamma distribution. Bootstrap support values were generated using 500 rapid bootstrap replicates.

**Table 1 pathogens-13-01009-t001:** A summary of low-pathogenic avian influenza viruses (LPAIVs) found in seal populations.

Subtype	Isolation	Year	Isolate	Closely Related Strain (Segment)	Similarity (%)
H7N7	Harbor seal in the Northeast United States	1979–1980	A/Seal/Mass/1/80 (H7N7)	A/mallard duck/ALB/224/1977 H7N5 (HA)	97.6
H4N5	Harbor seal in the Northeast United States	1982–1983	A/seal/Massachussetts/133/1982 (H4N5)	A/blue-winged_teal/ALB/580/1979 H4N2 (HA)	91.89
				A/mallard_duck/ALB/291/1977 H4N1 (HA)	91.54
H3N3	Seals in Massachusetts, United States	1991–1992	H3N3: A/seal/MA/3984/1992, A/seal/MA/3911/1992, A/seal/Massachusetts/3911/1992	A/mallard_duck/New_York/157/1986 H3N6 (HA)	98–98.8
				A/mallard/Ohio/48/1986 H3N2 (HA)	98–98.8
				A/mallard/Ohio/264/1986 H3N8 (HA)	98–98.8
				A/mallard/Alberta/195/1989 H7N3 (NA)	99.40
				A/mallard/Alberta/353/1988 H2N3 (NA)	99.40
H4N6	Seals in Massachusetts, United States	1991–1992	N/A	N/A	
	Caspian seal, Russia	2002	A/Caspian_seal/Russia/1884/2002	A/duck/South Africa/1233A/2004 H4N8 (HA)	98.30
				A/duck/Nanjing/1102/2010 H4N8 (HA)	97.90
				A/mallard duck/Netherlands/7/2006 H10N6 (NA)	98.80
	Caspian seal, Russia	2012	A/Caspian_seal/Russia/T1/2012	A/duck/South Africa/1233A/2004 H4N8 (HA)	98.40
				A/duck/Nanjing/1102/2010 H4N8 (HA)	98
				A/mallard duck/Netherlands/7/2006 H10N6 (NA)	98.8
H1N1	Elephant seals in California, United States	2010	A/Elephant_seal/California/1/2010, A/elephant_seal/California/2/2010	A/Singapore/GP3733/2009 H1N1 (HA), A/Malaysia/2082543/2009 H1N1 (HA), A/Texas/JMS358/2009 H1N1 (HA)	99.8
				A/Singapore/DMS7/2009 H1N1 (NA), A/Scotland/Edinburgh_23151/2009 H1N1 (NA)	99.5–99.6
		2016	A/elephant seal/California/UCD10NESmdck/2016 H1N1	A/Sao Gabriel/LACENRS-1626/2009 H1N1 (HA), A/Singapore/GP3733/2009 H1N1 (HA)	99.7
				A/Bretagne/7608/2009 H1N1 (NA), A/Ontario/35273/2009 H1N1 (NA)	99.3
		2018	A/Elephant seal/California/SL08B72H1/2018 H1N1	A/Sao Gabriel/LACENRS-1626/2009 H1N1 (HA), A/Singapore/GP3733/2009 H1N1 (HA)	99.7
				A/Florida/102/2018 H1N1 (NA)	99.9
		2019	A/elephant seal/California/ES4506NS/2019 H1N1	A/Florida/102/2018 H1N1 (HA)	99.9
				A/Bretagne/7608/2009 H1N1 (NA), A/Ontario/35273/2009 H1N1	99.4
H3N8	Harbor seals in Maine, New Hampshire, and Massachusetts, United States	2011	A/harbor seal/New Hampshire/179629/2011, A/harbor seal/Massachusetts/1/2011	A/blue-winged teal/New Brunswick/00283/2010 H3N8 (HA)	98.4
				A/Blue-winged Teal/New Mexico/A00327347/2007 H3N8 (HA)	98.4
				A/mallard/Minnesota/AI09-2749/2009 H6N8 (NA)	99
	Gray seal in the United Kingdom	2017	A/grey seal/England/027661/2017	A/barnacle goose/Netherlands/2/2014 H3N6 (HA)	98.5
				A/duck/Moscow/4661/2011 H3N8 (HA)	98.4
				A/greylag goose/Iceland/5751/2016 H6N8 (NA)	99.2
	Harbor seals in New Hampshire, United States	2018	A/harbor_seal/New_Hampshire/M11t72H3/2018, A/harbor seal/New Hampshire/M15t72H3/2018, A/harbor seal/New Hampshire/SL06B48H3/2018, A/harbor seal/New Hampshire/SL06B72H3/2018 *	A/blue-winged teal/New Brunswick/00283/2010 H3N8 (HA)	98.4
				A/mallard/Ohio/11OS2150/2011 H3N1 (HA)	98.3
				A/mallard/Minnesota/AI09-2749/2009 H6N8 (HA)	99.2
H10N7	Seals in Germany, Denmark, Sweden, the Netherland	2014	Multiple European strains	A/avian/Israel/543/2008 H10N7 (HA)	96.9–97.5
				A/Anas_platyrhynchos/Belgium/11798_H189445/2016 H10N5 (HA)	95.9–96.8
			A/harbour_seal/Germany/1/2014|HA||A_/_H10N7|EPI_ISL_170566	A/mallard duck/Netherlands/31/2013 H10N7 (HA)	98.4
			A/harbor_seal/NL/PV14-221_Ts/2015|HA||A_/_H10N7|EPI_ISL_202005		98.2
			A/harbor_seal/Denmark/14-5061-1lu/2014-07|HA||A_/_H10N7|EPI_ISL_166244		98.6
			A/Seal/Sweden/SVA0824/2014_/H10N7|HA||A_/_H10N7|EPI_ISL_167906		98.8
			A/harbour_seal/Germany/1/2014|HA||A_/_H10N7|EPI_ISL_170566	A/mallard duck/Netherlands/13/2013 H7N7 (NA)	99.6
			A/harbor_seal/NL/PV14-221_Ts/2015|HA||A_/_H10N7|EPI_ISL_202005		99.5
			A/harbor_seal/Denmark/14-5061-1lu/2014-07|HA||A_/_H10N7|EPI_ISL_166244		99.6
			A/Seal/Sweden/SVA0824/2014_/H10N7|HA||A_/_H10N7|EPI_ISL_167906	A/duck/Republic of Georgia/2/2010 H10N7 (NA)	99
	Harbor seal in British Columbia, Canada	2021	A/harbor_seal/British_Colombia/OTH-52-1/2021(H10N7)	A/duck/Bangladesh/24035/2014 H10N1 (HA)	95.8
				A/duck/Mongolia/97/2014 H10N6 (HA)	95.6
				A/duck/Tainan/16WB3373-26-30/2016 H7N7 (NA)	97.1

* Isolate used for analysis.

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
