# Peer review of "In Silico Genomic Analysis of Avian Influenza Viruses Isolated From Marine Seal Colonies"

_pathogens, 2024, doi:10.3390/pathogens13111009_

Round 1
Reviewer 1 Report
Comments and Suggestions for Authors
Abstract: More information are very useful to be added in the abstract ( type of LPAI , dates, location, species affected).
Line 4 : cased ( caused)
line 6 :H8Nx ( H5N8 clade 2.4.4.4b)
Page 3 : It will be more convenient for the reader if every type of LPAI is presented separately, like the authors did for H3N8,H10N7..
Page 15 : The first reported cases of highly pathogenic avian influenza viruses (HPAIV) in seals
( previously you mentioned H5N8 clade 2.4.4.4b not 2.3.4.4b ? very confusing ? the same confusion in the Summa
I suggest for the authors to summarize these results on a Map ,this will be very helpful for the reader.
Conclusion is needed
Author Response
Author`s reply: We would like to thank you for your feedback. The tile and main text was revised and proofread. Summary/abstract and data analysis section was added to the text. We also add some additional tables to suppl. material. We hope the manuscript is clearer to the reader in this version.
Reviewer 2 Report
Comments and Suggestions for Authors
The manuscript descripted by Klaudia Chrzastek and Darrell Kapczynski the genomic features of AIV in seals. The data in this manuscript showed the phylogenetic analysis of various viral genes of LPAIV and HPAIV, which has certain reference value. However, there are also some shortcoming in the manuscript.
1. Multiple paragraphs describe the timing of the discovery of AIV in seals, but the potential significance of these strains appearing at different time points is not clearly explained. I feel that more of the content is a listing of reported data, rather than elevating it to the significance of clinical aquaculture and public health.
2. At the end of the abstract, there is a lack of description of the significance and value of the research.
3. In Line 1 in second page, “avian influenza virus (AIV)” ? When the avian influenza virus first appears, use its full name and abbreviation, and when it reappears, use the abbreviation directly.
Author Response
We would like to thank you for your feedback. The title, main text was revised and proofread. Summary/abstract and data analysis section was added to the text. We also add some additional tables to suppl. material. We hope the manuscript is clearer for the reader in the current version.
Reviewer 3 Report
Comments and Suggestions for Authors
See attached file.

Round 2
Reviewer 2 Report
Comments and Suggestions for Authors
The author of this manuscript has made detailed revisions to address the issues raised by the reviewers. Also,the author provided reasonable explanations and supplements for some data in the manuscript. The author has made uniform revisions to the format of all references. I suggest accepting this manuscript。
Author Response
Thank you for your feedback. We revised the manuscript, add new table to the main document, and re-run phylogenetic analysis including more wild bird isolates. We believe that the article is more comprehensive in this form. Thank you
Reviewer 3 Report
Comments and Suggestions for Authors
See fiel attached.

Author Response
Please see below (attached).

Round 3
Reviewer 3 Report
Comments and Suggestions for Authors
Find, please, attached file.

Author Response
Dear Reviewer, thank you for your feedback. We implemented your suggestions in the main manuscript and supplementary file too. We also deleted one strain from the table as per request.
Thank you
Klaudia